# PAH-former: Transfer learning for efficient discovery of pulmonary arterial hypertension-associated genes

Toshinaru Kawakami[1,☉], Sosuke Hosokawa[1,2,☉], Masamichi Ito[1*],
Atsumasa Kurozumi[1], Ryohei Tanaka[1], Shun Minatsuki[1], Junichi Ishida[1],
Takayuki Isagawa[3,4], Satoshi Kodera[1*], Norihiko Takeda[1,3]

1 Department of Cardiovascular Medicine, Graduate School of Medicine, The University of Tokyo, Tokyo, Japan, 2 Department of Information and Communication Engineering, Graduate School of Information Science and Technology, The University of Tokyo, Tokyo, Japan, 3 Division of Bioconvergence, Center for Molecular Medicine, Jichi Medical University, Tochigi, Japan, 4 Data Science Center, Jichi Medical University, Tochigi, Japan

☉ These authors contributed equally to this work.
* mitou.tky@gmail.com (MI); koderasatoshi@gmail.com (SK)

## Abstract

### Background and Aims

Pulmonary arterial hypertension (PAH) is a severe disease with limited effective therapies, making the discovery of new therapeutic targets crucial. While single-cell RNA sequencing (sc-RNA seq) offers a powerful tool for this purpose, its application is hampered by the scarcity of patient samples. This study addresses the problem of how to efficiently identify novel, functionally relevant disease-associated genes from limited publicly available data.

### Methods

We employed transfer learning by fine-tuning Geneformer, a deep learning model, with public sc-RNA seq data from patients with PAH to create a specialized model called PAH-former. This model was used to perform *in silico* perturbation analysis to identify and rank candidate genes predicted to influence the disease state. For validation, we performed RNA interference-mediated knockdown of top novel candidate genes in human pulmonary artery endothelial cells and measured the expression of SRY-Box Transcription Factor 18 (*SOX18*), a signature gene of pulmonary arterial hypertension.

### Results

In silico perturbation analysis identified 134 candidate genes whose deletion was predicted to shift cells towards a disease phenotype. These included known disease-related genes as well as many novel ones. Subsequent *in vitro* validation

**Data availability statement:** The datasets generated and analyzed during the current study are available in the figshare repository (https://doi.org/10.6084/m9.figshare.31361932). The pretrained PAH-former models, including those trained on three distinct datasets, are publicly available via the Hugging Face Hub repository (https://huggingface.co/so298/PAH-former). Each model is published on a separate branch: "base_dataset," "add_hlca," and "add_other_data." Additionally, the source code for the PAH-former analyses is openly accessible through the GitHub repository (https://github.com/UTcardiology/PAH-former-analysis).

**Funding:** This work was supported by Cross-ministerial Strategic Innovation Promotion Program (SIP) on "Integrated Health Care System" Grant Number JPJ012425. There was no additional external funding received for this study. The funders had no role in study design, data collection and analysis, decision to publish, or preparation of the manuscript.

**Competing interests:** The authors have declared that no competing interests exist.

demonstrated that knockdown of the candidate genes resulted in a significant increase in the expression of *SOX18*.

## Conclusions

Our novel platform, PAH-former, provides a powerful and broadly applicable strategy for disease-related gene discovery. This approach enables the identification and validation of new candidate genes from limited data, promising to advance cell-specific mechanistic insights and accelerate therapeutic development for rare diseases like PAH. (248/300 words).

## Introduction

PAH is a severe disease characterized by progressive pulmonary vascular remodeling, which involves multiple cell types and leads to right heart failure and death [1]. However, effective treatments are lacking due to its rarity and the absence of accurate *in vivo* models mimicking human disease. Identifying drug targets is thus challenging, often requiring lengthy knockout animal validation. Moreover, understanding PAH fully requires cell-specific analysis to clarify each cell type's molecular roles. Therefore, innovative methods are essential to overcome these hurdles and discover effective PAH therapy targets. Notably, *SRY-Box Transcription Factor 18 (SOX18)*, a key transcription factor, has been identified as a signature gene whose expression is significantly upregulated in PAH, particularly in endothelial cells. Its involvement in angiogenesis and endothelial function highlights its importance in PAH pathogenesis, making it a valuable indicator of disease-like cellular states [2,3].

Although single-cell RNA sequencing (scRNA-seq) offers extraordinary resolution for dissecting complex diseases, its potential is often hindered by the limitations of conventional analytical methods. Conventional differential gene expression (DEG) analysis usually finds many related genes, but it is challenging to prove that they are the cause of the disease. This highlights the need for a more sophisticated approach, such as Geneformer, to identify truly functional disease-driving genes from complex transcriptomic data, particularly in rare diseases like PAH where sample acquisition is a significant bottleneck

Geneformer [4] is a foundational transformer model pre-trained on a large-scale corpus of single-cell transcriptomes to enable context-aware predictions in network biology. It was originally trained on approximately 30 million single-cell transcriptomes in June 2021, and later expanded to about 95 million transcriptomes in April 2024. Geneformer, with its ability to learn complex gene expression patterns and relationships from massive transcriptomic data in various conditions, offers an unbiased and robust approach to identify disease associated genes beyond DEG analysis. Although applications in understanding clonal pathologies such as cancer have been reported [5–7], there is no precedent for its use in elucidating complex systemic conditions like cardiopulmonary diseases.

In this study, to overcome these challenges and improve scRNA-seq data analysis quality, we built a novel platform based on Geneformer (PAH-former) and trained it based on public data of PAH. We also tested the effectiveness of addition of datasets in improving prediction accuracy and validated the established models by *in vitro* experiments. Our approach not only avoids the limitations of traditional DEG analysis-based methods but also demonstrates the broader applicability of Geneformer based fine-tuning as a powerful strategy for identifying disease associated genes. This study leverages our novel platform to identify and validate new disease associated genes of PAH, promising to advance our understanding of cell specific disease mechanisms and pave the way for novel therapeutic strategies.

## Materials and Methods

### Data collection and ethical statement

This study utilized publicly available datasets (GSE169471, GSE210248, GSE185479, and the Human Lung Cell Atlas) [2,8,9]. The data were accessed for research purposes from 1 June, 2024–30 October, 2024. The authors had no access to information that could identify individual participants during or after data collection.

For details of the materials and methods, please refer to the supplementary material.

### Model evaluation

To evaluate the impact of different training data compositions on performance, we fine-tuned three distinct models, designated A, B, and C. For consistent evaluation, a common test set was constructed from samples in GSE169471 that were not included in the training data (1 PAH sample and 2 control samples), consisting of 1,839 PAH cells and 2,780 control cells. Each model was trained for the binary classification task (PAH vs. control) using a different dataset partition:

Model A: Fine-tuned using only the GSE169471 dataset (6,514 PAH cells and 5,106 control cells).

Model B: Fine-tuned using the GSE169471 dataset combined with 10,000 control cells randomly sampled from the Human Lung Cell Atlas (HLCA) (6,514 PAH cells and 15,106 control cells).

Model C: Fine-tuned using a combined dataset comprising all three PAH-related datasets (GSE169471, GSE210248, and GSE185479) with 6,000 additional PAH cells randomly sampled, plus the control samples from the HLCA (12,514 PAH cells and 15,106 control cells).

### *In silico* perturbation

*In silico* perturbation was conducted on models A, B, and C, following the approach described previously in the Geneformer study. Briefly, this method perturbed the gene expression ranking to simulate gene inhibition or activation within single-cell transcriptomes. Genes targeted for perturbation were comprehensively selected from those expressed in both control and PAH samples. *In silico* deletion was simulated by removing targeted genes from the rank encoding, measuring perturbation effects via cosine similarity changes in both cell-level and gene-level embeddings. Conversely, *in silico* overexpression was simulated by moving the targeted genes to the top of the rank encoding, modeling the activation of these genes. Perturbations were executed using the test splits of datasets corresponding to each model. Two scenarios were explored: perturbations transitioning from control to PAH states, and vice versa, each involving both *in silico* deletion and overexpression strategies. Genes exhibiting a false positive rate below 0.05 and demonstrating a decreased cosine similarity toward the target state upon perturbation were considered promising candidates.

### Enrichment analysis

To identify potential driver mechanisms underlying PAH pathogenesis, we applied Metascape [10] to genes extracted by our PAH-former. The results of each perturbation output by Geneformer were ranked by the cosine shift towards the goal state ("Shift_to_goal_end") (largest first) and the False Discovery Rate (FDR) (smallest first). We defined genes

with a positive "Shift_to_goal_end" and an FDR < 0.05 as candidate disease-related genes. Candidate genes whose *in silico* deletion shifted the cell state towards PAH were 134 genes in total and we used all of them for enrichment analysis. Candidate genes of other directions were more than 200 genes. For enrichment analysis, we used the top 200 genes for enrichment analysis.

## Target Gene Knockdown using RNA Interference

To choose targets for experimental validation, we used a multi-step procedure. Top-ranked genes discovered by Control-to-PAH shift scores were manually curated, which included consultation with domain experts, to highlight physiologically viable but previously unknown candidates. Candidates were further screened for basal expression in human pulmonary artery endothelial cells (HPAECs) using the GSE228644 dataset [8], ensuring the feasibility of subsequent validation experiments.

HPAECs were commercially obtained (PromoCell, C-12241) and handled according to the provider's instructions. The cells were seeded in 96-well plates at a density of 2,400 cells/well in complete growth medium without antibiotics and incubated overnight. For *in vitro* knockdown experiment, the following siRNAs were purchased from Thermo Fisher Scientific: Silencer™ Select siRNAs for S100A6 (ID: s12418), TXNIP (ID: s12418), HSP90AA1 (ID: s6993), and MT2A (ID: s194629). siRNA was diluted in Opti-MEM I Reduced Serum Medium (Thermo Fisher Scientific, 31985070). For negative control, Silencer™ Select Negative Control No. 2 siRNA (Thermo Fisher Scientific, 4390846) stock solution was diluted with Opti-MEM I. Lipofectamine RNAiMAX reagent (Thermo Fisher Scientific, 13778150) was diluted with Opti-MEM. Equal volumes of diluted siRNA and diluted Lipofectamine RNAiMAX were combined and incubated at room temperature for 15 minutes to allow siRNA-Lipofectamine RNAiMAX complex formation. The volume of diluted Lipofectamine RNAiMAX solution was adjusted to use 0.2 μL of Lipofectamine RNAiMAX per well. The final concentration of each siRNA was 20 nM. Subsequently, siRNA-Lipofectamine RNAiMAX complexes were added to the cells. Cells were incubated with the complexes in a final volume of 120 μL per well at 37°C in a 5% $CO_2$ incubator for 48 hours post-transfection. Gene knockdown efficiency was evaluated at the indicated time points by quantitative PCR using QuantStudio 6 Flex Real-Time PCR System (Thermo Fisher Scientific). The mRNA expression levels of *SOX18*, as well as *VCAM1*, *ICAM1*, *ITGB1*, and *IL-6*, were evaluated for each condition. Data was normalized to the *GAPDH* expression level. The primer sequence used for the quantitative PCR analysis was provided in the Supplementary material.

## Statistical information

All quantitative data are presented as median with interquartile range (IQR). Statistical significance of differences between two groups was determined using a two-tailed Mann-Whitney *U* test. A *P* value of less than 0.05 was considered statistically significant. Specifically, for the RNA interference experiments, the mRNA expression levels of target genes (*S100A6, HSP90AA1, TXNIP, MT2A*) and *SOX18* in HPAECs following siRNA-mediated knockdown were compared against control siRNA-treated cells (Supplementary material). All statistical analyses were performed using GraphPad Prism version 10.4.2 (Dotmatics).

## Estimation of natural fluctuation threshold

To distinguish true biological signals from technical noise or natural biological variance, we performed a permutation-based analysis using the control samples from the validation dataset GSE228644. We randomly split the 1,307 healthy donor cells into two equal groups 100 times and calculated the $\log_2$ fold change ($\log_2$FC) for all genes in each iteration. From the resulting distribution of background $\log_2$ FC values, we defined the "natural fluctuation zone" as the range covering the 95th percentile. An observed $\log_2$FC in the disease group exceeding this threshold was considered a robust change beyond natural variance.

## Results

We conducted fine-tuning of Geneformer by public scRNA-seq analysis data to create "PAH-former", which can efficiently detect PAH associated genes (Fig 1A). PAH-former was trained using publicly available idiopathic pulmonary arterial hypertension (IPAH) datasets and can be utilized for various downstream analyses, such as cell type prediction and *in silico* perturbation. Single-cell data of IPAH is very limited and we primarily utilized data from GSE169471. First, we re-performed clustering and cell type annotation using the raw data from GSE169471 with CellTypist v2.0 [11]. As a result, we achieved clustering and cell type annotation comparable to the t-SNE presented in the original paper [2] (Fig 1B). We proceeded to map the cells, distinguishing between PAH and healthy groups. Our analysis revealed that each cluster contained cells from both control and PAH, aligning with the results presented in the original paper (Fig 1C). When we mapped the expression levels of *SOX18,* we similarly observed its selective expression in endothelial cells (Fig 1D).

## Training dataset setup

According to Geneformer's *in silico* perturbation of cardiomyocytes in the original paper, the training data included 93,589 cardiomyocytes (non-failing, n = 9; hypertrophic, n = 11; dilated, n = 9); the test data consisted of 39,006 cardiomyocytes

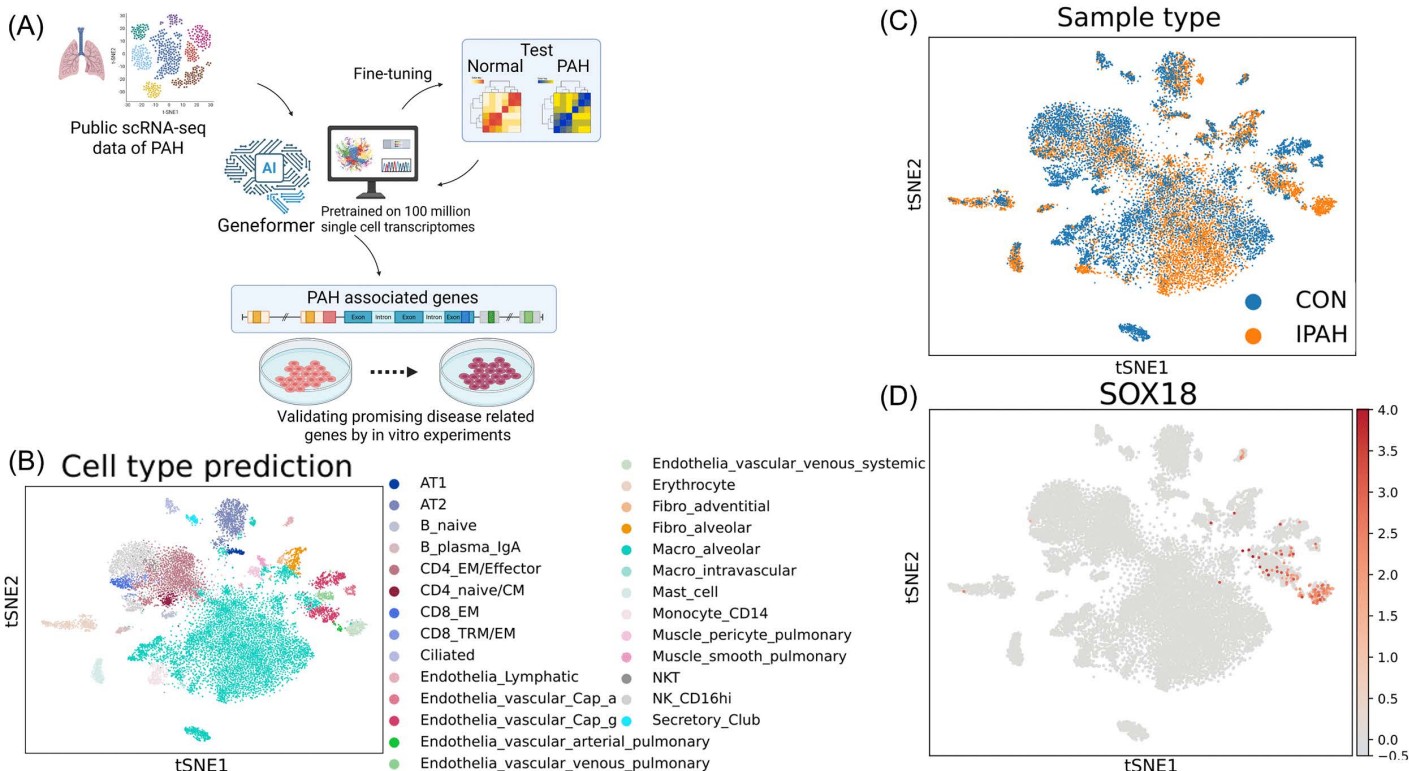

**Fig 1. Public single cell RNA-seq data of pulmonary artery hypertension lung and its reanalysis. (A)** Schematic of the PAH-former development pipeline. We created a new AI tool that understands the gene expression network of pulmonary arterial hypertension (PAH) by fine-tuning Geneformer, a transfer learning tool that has been trained on 1 billion single-cell analysis data, with publicly available PAH single-cell analysis data. The aim is to extract the genes that are involved in PAH. scRNA-seq, single-cell RNA sequencing; t-SNE, t-distributed stochastic neighbor embedding; PAH, pulmonary arterial hypertension. (B) t-SNE plot showing cell type prediction. Cell type annotation of the GSE169471 data was performed using CellTypist v2.0. t-SNE, t-distributed stochastic neighbor embedding. (C) t-SNE plot showing sample type distribution. CON, control; IPAH, idiopathic pulmonary artery hypertension; t-SNE, t-distributed stochastic neighbor embedding. (D) t-SNE plot visualizing *SOX18* expression levels. t-SNE, t-distributed stochastic neighbor embedding.

(non-failing, n = 4; hypertrophic, n = 4; dilated, n = 2) [4]. However, publicly available IPAH data is very limited, and its quality also varies, making it potentially difficult to secure a sufficient amount for fine-tuning of Genformer for IPAH. While using large datasets for fine-tuning carries a risk of overfitting, it has been reported that fine-tuning can be performed efficiently even with small datasets [12,13].

To optimize our model's ability to classify cells as either PAH or control, we compared three distinct training approaches (Models A, B, and C, detailed in Fig 2A). We found that the inclusion of large control data from Human Lung Cell Atlas (HLCA) significantly enhanced the model's accuracy and F1 score in cell classification of PAH or control. Fig 2A presents a table outlining the datasets used to train the three fine-tuning models evaluated in this study. Model A, which used only data from GSE169471, performed very poorly. Model B showed significant improvement with the addition of healthy control cells from the Human Lung Cell Atlas (HLCA). Model C, despite including other IPAH data, performed slightly worse

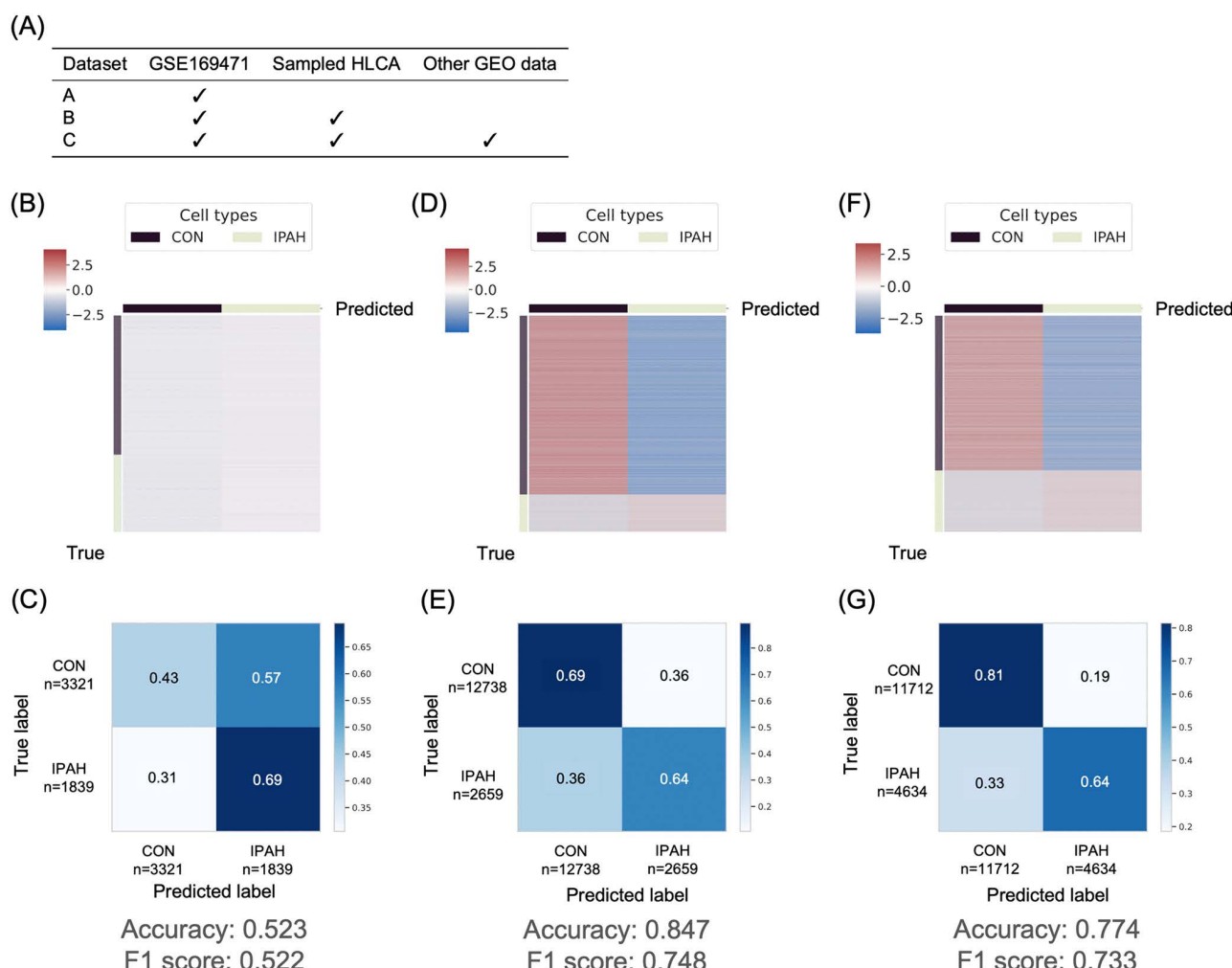

**Fig 2. Dataset selection for the fine-tuning of Geneformer. (A)** Datasets used for Geneformer model training. This table outlines the composition of training data for each model. HLCA, human Lung Cell Atlas. **(B, C)** Prediction likelihood heatmap for cell classification and confusion matrix of Model **A. (D, E)** Prediction likelihood heatmap for cell classification and confusion matrix of Model **B. (F, G)** Prediction likelihood heatmap for cell classification and confusion matrix of Model **C.** CON, control; IPAH, idiopathic pulmonary artery hypertension.

than Model B (Fig 2B-G). This observation suggests that for this specific task, the quality of control data from HLCA had a more significant positive impact than simply increasing the number of IPAH data.

### *In silico* perturbation analysis

To investigate the impact of specific genes on cell state in PAH, we performed in silico deletion and overexpression analyses using Geneformer fine-tuned by PAH scRNA-seq data (GSE169471) (Fig 3A). This analysis encompassed four distinct perturbation scenarios. As a result, we generated lists of putative disease associated genes for each of the four scenarios, which included genes previously implicated in the disease. Herein, we report the top 40 candidate genes identified for each scenario, along with the corresponding results of GO analysis of genes in each list (Fig 3B-E).

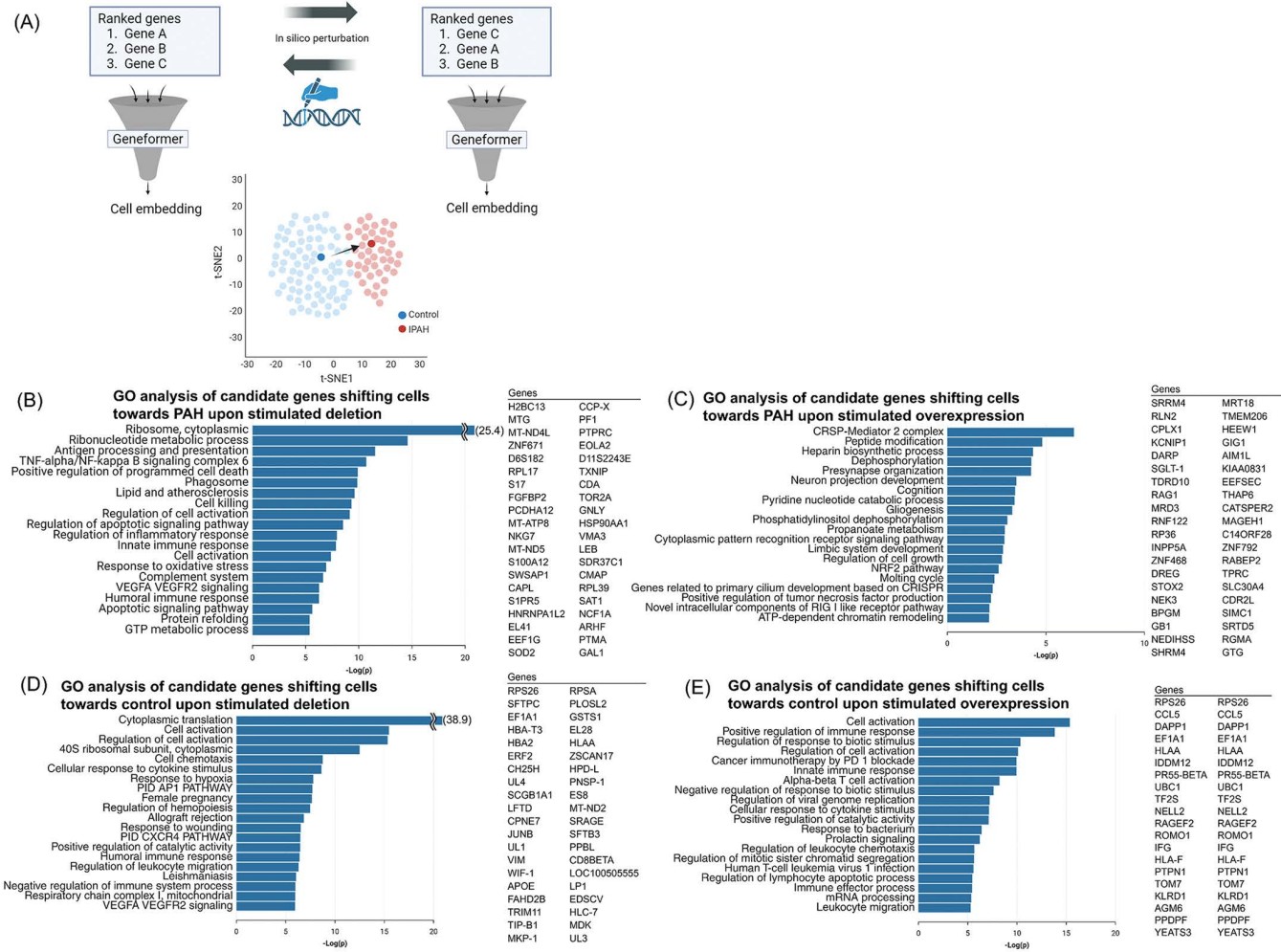

**Fig 3. *In silico* perturbation using PAH-former and extraction of disease-related genes.** (A) The workflow for *in silico* perturbation analysis using the fine-tuned Geneformer model (PAH-former). *In silico* manipulation (deletion or overexpression) results in shifts in cell embedding (representing cell state). IPAH, idiopathic pulmonary artery hypertension. (B-E) Gene Ontology (GO) analysis of candidate genes identified by *in silico* perturbation analysis by PAH-former and top 40 genes that shift cell embedding most for each of the four directions. (B) Gene Ontology (GO) analysis for candidate genes whose *in silico* deletion shifts the cell state towards IPAH. (C) Gene Ontology (GO) analysis for candidate genes whose *in silico* overexpression shifts the cell state towards control. (D) Gene Ontology (GO) analysis for candidate genes whose in silico overexpression shifts the cell state towards IPAH. (E) Gene Ontology (GO) analysis for candidate genes whose in silico deletion shifts the cell state towards control.

The number of candidate genes extracted by the fine-tuned Geneformer that shifts cell embedding from control to PAH state after in silico deletion were 134 (Supplementary table 1). Among the identified genes, while some were previously reported, the majority of them were novel. Previously reported genes included *HMGB2* (high-mobility group box 2). HMGB2 is upregulated in PAH, and it is mentioned as a significant contributor to the pathogenesis of pulmonary hypertension by promoting inflammation and vascular remodeling [14]. In addition, *SOD2* (superoxide dismutase 2) was also on the gene list, and its tissue specific, epigenetic downregulation initiates and sustains PAH by impairing redox signaling and promoting a proliferative, apoptosis-resistant pulmonary artery smooth muscle cell phenotype [15]. Notably, many genes in this list were not previously linked to pulmonary hypertension, suggesting them as novel candidates for exploring disease related molecular functions and pathways. Furthermore, enrichment analysis of the 134 genes highlighted the enrichment in TNF-α/NF-κB signaling, regulation of inflammatory response, oxidative stress response, and VEGFA/VEGFR2 signaling pathways. There was minimal overlap between the gene set identified by our fine-tuned Geneformer and that derived from the DEG analysis in the original article, with only two genes being common to both lists (Fig 4A, 4B). This demonstrates that PAH-former effectively found functionally significant disease pathways that were missed by conventional DEG analysis, highlighting the importance of using a context-aware deep learning model.

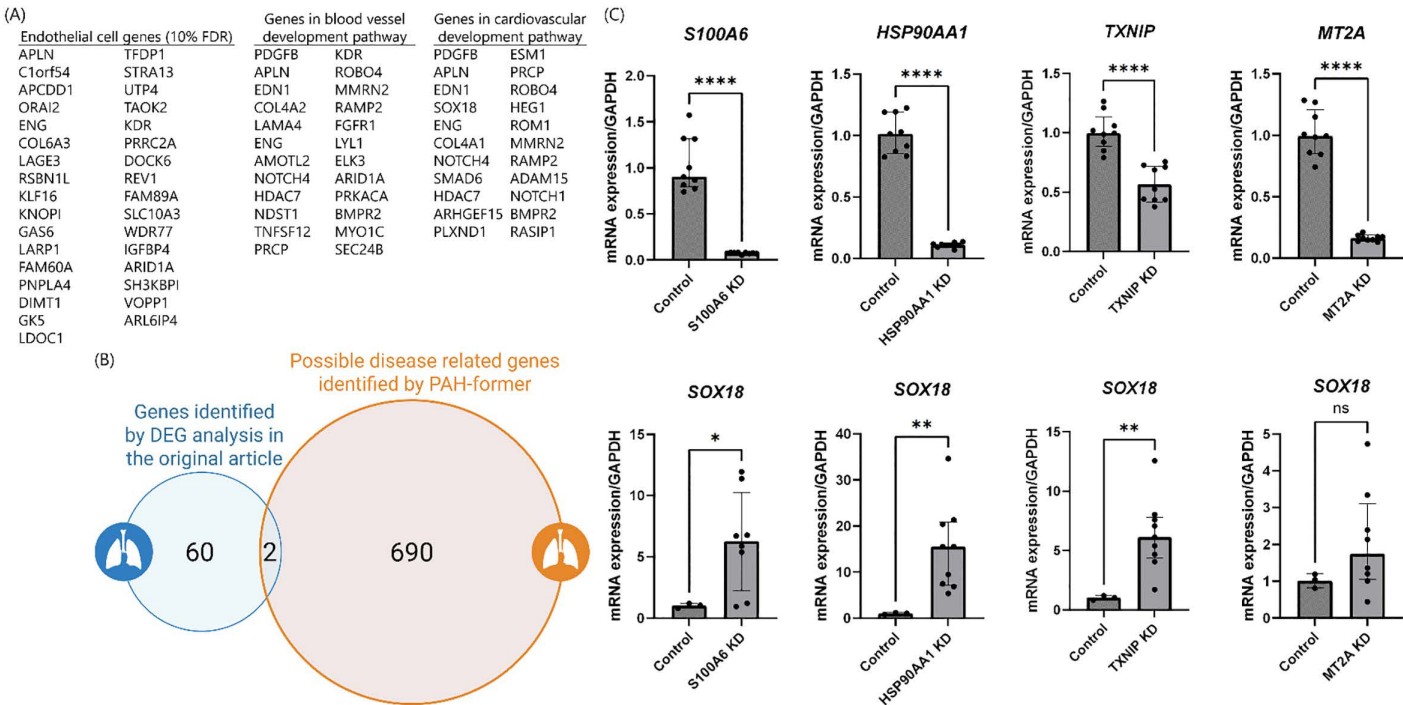

**Fig 4.** ***In Silico*** **perturbation in pulmonary endothelial cells using PAH-former and** ***in vitro*** **validation. (A)** Table showing partial lists of genes identified in endothelial cells by Differential Gene Expression (DEG) analysis (FDR < 10%) in the original article and genes enriched in the GO (gene ontology) pathways of blood vessel development pathway and cardiovascular development pathway. **(B)** Venn diagram comparing the number of candidate genes identified by Differential Gene Expression (DEG) analysis in the original article and the number of disease-associated candidate genes identified by PAH-former. PAH, pulmonary arterial hypertension. **(C)** Knockdown experiments of selected candidate genes (*S100A6, HSP90AA1, TXNIP,* and *MT2A*) using siRNA in Human Pulmonary Artery Endothelial Cells (HPAECs). Top panels show the knockdown efficiency of each target gene relative to control siRNA, displaying relative mRNA expression levels at 48 hours and indicating successful knockdown (data normalized to *GAPDH* expression, n = 9). Bottom panels show the relative mRNA expression levels of *SOX18* at 48 hours after knockdown of each candidate gene compared to control siRNA (data normalized to *GAPDH* expression, n = 9). Graphs are presented as median with interquartile range (IQR). Statistical significance is indicated as: * $p < 0.05$, ** $p < 0.01$, **** $p < 0.0001$, ns: not significant.

## Target Gene Knockdown

Among the genes listed as PAH-related, we picked up four genes (*S100 Calcium Binding Protein A6 [S100A6], Heat Shock Protein 90 Alpha Family Class A Member 1 [HSP90AA1], Thioredoxin Interacting Protein [TXNIP],* and *Metallothionein 2A [MT2A]*), which have not been previously reported in association with PAH.

In the original paper on which our dataset is based, several transcription factors, particularly *SOX18*, were found to be upregulated and implicated in regulating the PAH endothelial cell transcriptome. *SOX18* is also reported to increase the expression of nicotinamide phosphoribosyltransferase (NAMPT) and is shown to be involved in the pathophysiology of PAH via NAMPT [3]. Therefore, in order to determine whether the cellular state was approaching a PAH-like state after knockdown of candidate genes found by PAH-former, we chose to compare mRNA expression of *SOX18*.

We performed a knockdown experiment of the four candidate genes using RNA interference with HPAECs. For each gene targeted, the knockdown was successful. Interestingly, the knockdown of three of these four genes resulted in a significant increase in *SOX18* expression 48 hours after siRNA transfection (Fig 4C), ensuring the validity of the candidates extracted by PAH-former. To validate our findings, we utilized an independent scRNA-seq dataset (GSE228644). This analysis confirmed a significant upregulation of *SOX18* in PAH endothelial cells compared to donors ($\log_2 FC = 0.85$). To address the potential for natural variation, we established a natural fluctuation threshold ($\log_2 FC \pm 0.68$) based on a permutation analysis of healthy donor cells. The upregulation of *SOX18* significantly exceeded this noise threshold (1.24-fold above the threshold limit), confirming that the observed expression change represents a distinct pathological feature rather than intrinsic biological noise.

To further validate the phenotypic shift towards a PAH-like state, we additionally examined the expression of adhesion molecules (*VCAM1, ICAM1* and *ITGB1*) and the pro-inflammatory cytokine *IL-6*, which are known to be upregulated in PAH endothelial cells [16–18]. Consistent with the *SOX18* results, knockdown of *HSP90AA1* and *TXNIP* led to a significant upregulation of *VCAM1, ITGB1*, and *IL-6*. *S100A6* knockdown also resulted in significant increases in *ICAM1* and *ITGB1* expression (Fig. S2 in S1 File). These broad changes in disease-associated markers strongly support the functional relevance of the candidates identified by PAH-former.

## Discussion

Many different genes and pathways are involved in the development of PAH. For our study's validation [19–24], we focused on the transcription factor *SOX18*, as its expression is a key feature of the PAH endothelial cell phenotype. Given that *SOX18* is significantly upregulated in IPAH and plays a critical role in vascular remodeling [25–27], its modulation serves as a robust indicator of a shift towards or away from a disease-like state. This context is crucial for interpreting our finding that the knockdown of several novel candidate genes directly impacted *SOX18* expression.

The candidate genes were *S100A6, HSP90AA1, TXNIP* and *MT2A*. Although these genes have not been previously reported in association with PAH, their known biological functions offer plausible connections to disease mechanisms. For instance, *S100A6* (also called *Calcyclin, Cacy*) is a Ca2 + -binding protein involved in cell proliferation and stress, with previous research suggesting its role in regulating antiproliferative pathways and potential interactions with proteins like Calcyclin-Binding Protein and Siah-1 Interacting Protein (CacyBP/SIP) in PAH models, hypothetically impacting vascular remodeling [28–30]. *HSP90AA1* plays a crucial role in maintenance of endothelial nitric-oxide synthase (eNOS) dimer stability in pulmonary arterial endothelial cells and is an upregulated immune-related gene in PAH, implying its involvement in endothelial dysfunction and inflammatory processes [31,32]. *TXNIP* mediates oxidative stress by inhibiting thioredoxin activity, a system linked to PAH progression, suggesting that its knockdown could enhance pro-PAH conditions [33–36]. Lastly, MT2A, a member of the metallothionein family, possesses antioxidant properties and is elevated in PAH patients, indicating its potential role as a biomarker and a defense against oxidative stress; its deletion might compromise this protection, leading to PAH-like pathology [37].

Although these genes have entirely distinct functions, the common upregulation of *SOX18* observed upon their knock-down in HPAECs is particularly noteworthy. This consistency across multiple novel genes suggests that our PAH-former approach successfully identified candidates that influence a key PAH-associated cellular phenotype. The induction of a PAH-like phenotype was confirmed not only by *SOX18* upregulation but also by the increased expression of other key disease markers such as *VCAM1* and *IL-6* (Fig. S2 in S1 File), reinforcing the potential of these genes as drivers of the disease state. However, the extent of upregulation was variable across the tested markers. This is likely because our current model predicts global state transitions without explicitly describing the specific molecular mechanisms involved. Consequently, certain downstream pathways may not be fully activated. In future studies, we aim to refine the algorithm to better identify which specific biological pathways are mechanistically driven by the predicted gene perturbations. In this way, fine-tuning Geneformer using the PAH dataset and subsequently conducting *in silico* perturbation analysis enabled us to identify a significant number of previously unknown disease-associated genes in PAH. Our approach provides a significant benefit over conventional DEG analysis by finding genes that causally affect cellular states, rather than simply indicating changes in expression levels. This ability to pinpoint functionally relevant genes, even from limited patient samples, demonstrates the unique utility of a transfer learning framework like PAH-former. Furthermore, to benchmark our model against established clinical targets, we performed *in silico* perturbation analyses on genes targeted by current PAH therapies, such as *Phosphodiesterase 5A* (*PDE5A*), *Guanylate Cyclase 1 Soluble Subunit Alpha 3* (*GUCY1A3*), *Endothelin Receptor Type A* (*EDNRA*), and *Prostaglandin I2 Receptor* (*PTGIR*). Interestingly, perturbation of these individual genes did not induce a significant global shift in cell embedding towards the healthy state. While these targets are clinically effective for vasodilation and symptom management, modulating their expression alone may not be sufficient to reverse the complex, global transcriptomic dysregulation characteristic of the PAH endothelial phenotype. In contrast, when we focused on the *in silico* deletion analysis simulating the reversion from the PAH state to the control phenotype—which simulates the therapeutic inhibition of pathogenic genes—our model successfully prioritized major upstream regulators and signaling nodes, such as the Activator Protein 1 (*AP-1*) complex, Transforming Growth Factor Beta (*TGF-β*), and Secreted Protein Acidic and Cysteine Rich (*SPARC*) which are well-known key factors involved in the pathophysiology of PAH [38]. This suggests that PAH-former is particularly effective at identifying fundamental drivers required to normalize the global cellular state. While further validation is required, this approach offers a comprehensive means to explore therapeutic targets and has the potential to enhance the efficiency of fundamental experiments, reducing the effort and cost required for molecular function experiments. Moreover, the Geneformer and public database combination (fine-tuning) represents a promising new platform applicable to a wide range of diseases, particularly rare diseases where patient samples are scarce and the underlying pathological mechanisms are poorly understood. We believe it can greatly accelerate the advancement of our understanding of disease molecular mechanisms.

Our *in silico* overexpression analysis also identified promising candidates, such as Ephrin-B2 (*EFNB2*) and Repulsive Guidance Molecule BMP Co-Receptor A (*RGMA*), which are known to play critical roles in vascular homeostasis [39] and BMP signaling. Although we prioritized the experimental validation of knockdown targets in this study, the identification of these established markers suggests that our model captures bidirectional regulatory dynamics. Validation of these overexpression candidates is planned for future investigations.

While our platform demonstrated promising capabilities, several limitations warrant consideration. First, the gene outputs are inherently dataset dependent. Different datasets, even those examining similar biological contexts, may exhibit variations in gene expression profiles, potentially leading to discrepancies in the identified key genes or pathways. Second, the fine-tuning process of our model is sensitive to the choice of fine-tuning datasets and hyperparameters, such as the learning rate. Variations in these parameters can lead to different model outputs and potentially affect the robustness of our findings. Third, our study's focus on SOX18 mRNA expression as the primary validation metric is a limitation. While we have analyzed the expression levels of *SOX18*, we have not yet validated whether these expression changes are directly linked to corresponding changes in cellular phenotypes relevant to PAH. Finally, we must acknowledge the

multifactorial nature of PAH. Our gene-centric approach, while informative, may not fully capture the complexity of PAH, which likely involves intricate interactions of multiple genetic and environmental factors beyond single gene mutations, as well as cell-cell interaction change within the organs.

In conclusion, our novel Geneformer-based fine-tuning platform provides a powerful and broadly applicable strategy for disease-related gene discovery. This approach enables the identification and validation of new candidate genes, promising to advance cell-specific mechanistic insights and efficient therapeutic development for PAH.

## Suporting information

**S1 File. This file contains Supplementary Figures S1 and S2, along with detailed Supplementary Methods for the model architecture and experimental procedures. This Excel file contains the comprehensive lists of candidate genes identified by the in silico perturbation analysis.**
(ZIP)

## Acknowledgments

We would like to thank Yukiko Kaneko for her technical assistance.

## Author contributions

**Conceptualization:** Masamichi Ito.

**Data curation:** Sosuke Hosokawa.

**Funding acquisition:** Satoshi Kodera.

**Investigation:** Takayuki Isagawa.

**Methodology:** Sosuke Hosokawa.

**Resources:** Satoshi Kodera.

**Software:** Sosuke Hosokawa.

**Supervision:** Masamichi Ito, Satoshi Kodera, Norihiko Takeda.

**Validation:** Toshinaru Kawakami.

**Writing – original draft:** Toshinaru Kawakami, Sosuke Hosokawa.

**Writing – review & editing:** Toshinaru Kawakami, Masamichi Ito, Atsumasa Kurozumi, Ryohei Tanaka, Shun Minatsuki, Junichi Ishida, Takayuki Isagawa, Satoshi Kodera, Norihiko Takeda.

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
