## [Decision Letter · Decision Letter 0]

6 Jan 2026

Dear Dr. Ito,

Thank you for submitting your manuscript to PLOS ONE. After careful consideration, we feel that it has merit but does not fully meet PLOS ONE’s publication criteria as it currently stands. Therefore, we invite you to submit a revised version of the manuscript that addresses the points raised during the review process.

We look forward to receiving your revised manuscript.

Kind regards,

Nanako Kawaguchi

Academic Editor

PLOS One

**Journal Requirements:**

1. When submitting your revision, we need you to address these additional requirements. Please ensure that your manuscript meets PLOS ONE's style requirements, including those for file naming. The PLOS ONE style templates can be found at https://journals.plos.org/plosone/s/file?id=wjVg/PLOSOne_formatting_sample_main_body.pdf and https://journals.plos.org/plosone/s/file?id=ba62/PLOSOne_formatting_sample_title_authors_affiliations.pdf 2. We note that the grant information you provided in the ‘Funding Information’ and ‘Financial Disclosure’ sections do not match.  When you resubmit, please ensure that you provide the correct grant numbers for the awards you received for your study in the ‘Funding Information’ section. 3. Thank you for stating in your Funding Statement: This work was supported by Cross-ministerial Strategic Innovation Promotion Program (SIP) on “Integrated Health Care System” Grant Number JPJ012425.  Please provide an amended statement that declares *all* the funding or sources of support (whether external or internal to your organization) received during this study, as detailed online in our guide for authors at http://journals.plos.org/plosone/s/submit-now.  Please also include the statement “There was no additional external funding received for this study.” in your updated Funding Statement. Please include your amended Funding Statement within your cover letter. We will change the online submission form on your behalf. 4. Thank you for stating the following financial disclosure: This work was supported by Cross-ministerial Strategic Innovation Promotion Program (SIP) on “Integrated Health Care System” Grant Number JPJ012425.   Please state what role the funders took in the study.  If the funders had no role, please state: "The funders had no role in study design, data collection and analysis, decision to publish, or preparation of the manuscript." If this statement is not correct you must amend it as needed. Please include this amended Role of Funder statement in your cover letter; we will change the online submission form on your behalf. 5. Thank you for stating the following in the Acknowledgments Section of your manuscript: This work was supported by Cross-ministerial Strategic Innovation Promotion Program (SIP) on “Integrated Health Care System” Grant Number JPJ012425.We would like to thank Yukiko Kaneko for her technical assistance. We note that you have provided funding information that is not currently declared in your Funding Statement. However, funding information should not appear in the Acknowledgments section or other areas of your manuscript. We will only publish funding information present in the Funding Statement section of the online submission form. Please remove any funding-related text from the manuscript and let us know how you would like to update your Funding Statement. Currently, your Funding Statement reads as follows: This work was supported by Cross-ministerial Strategic Innovation Promotion Program (SIP) on “Integrated Health Care System” Grant Number JPJ012425. Please include your amended statements within your cover letter; we will change the online submission form on your behalf. 6. Please note that your Data Availability Statement is currently missing the direct link to access each database. If your manuscript is accepted for publication, you will be asked to provide these details on a very short timeline. We therefore suggest that you provide this information now, though we will not hold up the peer review process if you are unable. 7. When completing the data availability statement of the submission form, you indicated that you will make your data available on acceptance. We strongly recommend all authors decide on a data sharing plan before acceptance, as the process can be lengthy and hold up publication timelines. Please note that, though access restrictions are acceptable now, your entire data will need to be made freely accessible if your manuscript is accepted for publication. This policy applies to all data except where public deposition would breach compliance with the protocol approved by your research ethics board. If you are unable to adhere to our open data policy, please kindly revise your statement to explain your reasoning and we will seek the editor's input on an exemption. Please be assured that, once you have provided your new statement, the assessment of your exemption will not hold up the peer review process. 8. Please upload a new copy of Figures 1 – 4, as the detail is not clear. Please follow the link for more information:  https://journals.plos.org/plosone/s/figures 9. Please include captions for your Supporting Information files at the end of your manuscript, and update any in-text citations to match accordingly. Please see our Supporting Information guidelines for more information: http://journals.plos.org/plosone/s/supporting-information. 10. If the reviewer comments include a recommendation to cite specific previously published works, please review and evaluate these publications to determine whether they are relevant and should be cited. There is no requirement to cite these works unless the editor has indicated otherwise. 

**Additional Editor Comments:**

The ms is overall interesting and informative. However, my opinion, the authors should validate the results of Sox18 comparing PAH endotheilal cells and control ECs.

Reviewers' comments:

**Comments to the Author**

1. Is the manuscript technically sound, and do the data support the conclusions?

Reviewer #1: Partly

Reviewer #2: Partly

2. Has the statistical analysis been performed appropriately and rigorously?

Reviewer #1: Yes

Reviewer #2: Yes

3. Have the authors made all data underlying the findings in their manuscript fully available?

Reviewer #1: Yes

Reviewer #2: Yes

4. Is the manuscript presented in an intelligible fashion and written in standard English?

Reviewer #1: Yes

Reviewer #2: Yes

**Reviewer #1:** Thanks for allowing me to participate in ths review. The paper is well written.

The used transfer learning W by fine-tuning Geneformer, a deep learningmodel, with public sc-RNA seq data from patients with PAH to create a specialized model called PAH-former. This model was used to perform in silico perturbation analysis to identify and rank candidate genes predicted to influence the disease state.

For validation, we performed RNA interference-mediated knockdown of top novel candidate genes in human pulmonary artery endothelial cells and measured and compared them with the expression of SRY-Box Transcription Factor 18 (SOX18), a gene of pulmonary arterial hypertension. The authors then trained the model based on public data of PAH. We also tested the effectiveness of addition of datasets in improving prediction accuracy and validated the established models by in vitro experiments.

Major critique- The authors need to test for other genes associated with PAH other than SOX 18. They also should have prespecified thresholds for gene regulation based on natural fluctuations and variance in normal lungs.

Major Critique: There are many genes now identified perhaps the authors should compare vs genes with known targeted therapeutics.

The discussion is well organized and addresses some of the limitations.

Major Critique: the figures are not of high quality; some of the data is illegible.

**Reviewer #2:**  In this work, the authors implemented a transfer learning framework for identifying candidate genes associated with pulmonary arterial hypertension (PAH). Specifically, the authors fine-tuned the transformed-based Geneformer model (PAH-former) on single-cell RNA-seq data from PAH and performed in silico perturbation analysis to prioritize gene candidates predictive of the phenotype state.

Major Comments:

• For the target gene knockdown analysis section, the authors mentioned picking four genes for further validation. It is not clear what the rationale was behind choosing these genes. Is it data-driven or biology driven? This needs to be clarified or cleared up in the manuscript.

• Also, while the authors performed experiments associated with both in silico deletion and overexpression analysis using the PAH-former, the candidate genes were picked from the 134 gene list coming out of the in silico deletion analysis only. What is the reason behind it? Did the authors explore any candidates from the overexpression analysis?

• The data partitions for fine-tuning experiments seem to be a bit off. Specifically for model B, looks like the authors used a total of 106 controls + 2 IPAH samples for training while using 1 IPAH sample + 7 controls. Did the authors try using similar IPAH:controls ratio for both training and testing partitions?

• Also, what are the cell counts in these partitions? It might be relevant to include those in the manuscript.

• In the results section, the authors compared the disease related genes identified by PAH-former with DEGs from the previous article. In the 690 genes identified specifically by the PAH-former, were there any functional terms (pathways or processes) that the relevant to PAH or pulmonary disorders in general? If that is the case, it could be another good justification for using such complex models over the traditional DEG analysis.

Minor comments

• Looks like the authors used the short form ‘HPAECs’ before it was defined for the first time in the manuscript (the full-form was mentioned in a later section of the manuscript). This needs to be fixed.

• The word ‘which’ was misspelled (‘whcih’) in line 246 in the ‘Target Gene Knockdown’ section. Please correct this.

• From a reproducibility perspective, making the scripts used for fine-tuning the Geneformer model can be very useful. The authors should consider sharing their work.

**Do you want your identity to be public for this peer review?** For information about this choice, including consent withdrawal, please see our Privacy Policy

Reviewer #1: No

Reviewer #2: **Yes:** Sudhir Ghandikota

---

## [Author Response · Author response to Decision Letter 1]

2 Feb 2026

We thank the editor and all reviewers for their insightful remarks and suggestions regarding our manuscript. We also appreciate the opportunity to revise and resubmit the manuscript.

Based on the editor and the reviewers’ comments (in blue), we have performed additional experiments and revised the manuscript accordingly. Our changes are highlighted in red in this file and in yellow in the revised manuscript file.

Comments from the Editor

The ms is overall interesting and informative. However, my opinion, the authors should validate the results of Sox18 comparing PAH endotheilal cells and control ECs.

Response:

We fully agree that validating the results in clinical samples is crucial. To address this concern and robustly validate our findings, we utilized an independent external single-cell RNA sequencing dataset (GSE228644) (1) comprising PAH and healthy donor lung samples. The analysis revealed that SOX18 expression was significantly upregulated in PAH endothelial cells, showing an approximately 2-fold increase compared to healthy controls (Fig. S1A, B).

Furthermore, to ensure this upregulation represents a true biological signal rather than technical noise, we performed a permutation analysis, as detailed in our response to Reviewer 1. This analysis demonstrated that the observed SOX18 upregulation significantly exceeds the natural fluctuation threshold defined by random sampling of healthy donor cells. These external validation results strongly support our original findings and confirm the pivotal role of SOX18 in PAH.

We have included the figure below as Figure S1 in the revised manuscript. We address this in further detail in our response to Reviewer 1.

* : P < 0.05

Comments from Reviewer #1

Major critique- The authors need to test for other genes associated with PAH other than SOX 18. They also should have prespecified thresholds for gene regulation based on natural fluctuations and variance in normal lungs.

Response:

[The first half]

To address this comment, we focused on the following four genes that have been consistently reported to be highly expressed in PAH endothelial cells in previous studies: Vascular Cell Adhesion Molecule-1 (VCAM1), Intercellular Adhesion Molecule-1 (ICAM1), Interleukin-6 (IL-6) , and Integrin beta-1 (ITGB1).

VCAM1 and ICAM1 are proinflammatory adhesion molecules markedly elevated in the pulmonary vascular endothelium of IPAH patients and in cultured ECs from IPAH patients (2). It has been shown that the beta1 subunit, encoded by the ITGB1 gene, is expressed at significantly higher levels compared to other beta subunits in both pulmonary artery smooth muscle cells (PASMCs) and endothelial cells (PAECs) of PAH (3). It is well established that elevated serum IL-6 levels correlate with poor prognosis in PAH patients. In murine hypoxia-induced pulmonary hypertension lungs, IL-6 expression is markedly upregulated in pulmonary arteriolar endothelial and smooth muscle cells (4).

We performed knockdown (KD) in HPAECs targeting S100A6, HSP90AA1, and TXNIP, as these three genes were found to significantly upregulate SOX18 expression in our study. Samples were collected 48 hours post-transfection, consistent with our previous experiments. We then verified whether the expression levels of the prior four PAH-associated genes were elevated in the KD group compared to the control group.

Experimental results showed a significant increase in VCAM1 and IL-6 levels in the HSP90AA1 and IL-6 KD groups compared to the control group. The extent of this rise was more pronounced for VCAM1. Regarding ITGB1, an increase was observed across all three KD groups. While ICAM1 showed a significant increase only in the S100A6 KD group, the magnitude of the increase was small.

Collectively, these results indicate that the knockdown of candidates identified by PAH-former successfully induced the upregulation of other key PAH-associated genes, driving the cells toward a disease-like phenotype. However, we acknowledge that the response was variable, with some markers showing only mild or non-significant changes. This is likely because our current model predicts global state transitions but does not explicitly describe the specific molecular mechanisms leading to disease formation. We recognize this as an area for future development, and we aim to refine the algorithm to incorporate such mechanistic insights in subsequent studies

Based on the above experimental results and discussions, the paper was revised as follows.

・The qPCR results of HPAECs KD experiment were added as Fig. S2

The following description, corresponding legends and methods have been added to the text.

・The following points were added to Results (Target gene knockdown section).

“To further validate the phenotypic shift towards a PAH-like state, we additionally examined the expression of adhesion molecules (VCAM1, ICAM1 and ITGB1) and the pro-inflammatory cytokine IL-6, which are known to be upregulated in PAH endothelial cells (reference). Consistent with the SOX18 results, knockdown of HSP90AA1 and TXNIP led to a significant upregulation of VCAM1, ITGB1, and IL-6. S100A6 knockdown also resulted in significant increases in ICAM1 and ITGB1 expression (Fig. S2). These broad changes in disease-associated markers strongly support the functional relevance of the candidates identified by PAH-former.” (L.294-301)

・The following points were added to Discussion.

“The induction of a PAH-like phenotype was confirmed not only by SOX18 upregulation but also by the increased expression of other key disease markers such as VCAM1 and IL-6 (Fig. S2), reinforcing the potential of these genes as drivers of the disease state. However, the extent of upregulation was variable across the tested markers. This is likely because our current model predicts global state transitions without explicitly describing the specific molecular mechanisms involved. Consequently, certain downstream pathways may not be fully activated. In future studies, we aim to refine the algorithm to better identify which specific biological pathways are mechanistically driven by the predicted gene perturbations.” (L.332-340)

“The primers used for qPCR were as follows:

VCAM1, AGCACCACAGGCTCTTTTCC, TTGACTGTGATCGGCTTCCC

ICAM1, AGCTTCGTGTCCTGTATGGC, CTGGCACATTGGAGTCTGCT

IL-6, GAAAGTGGCTATGCAGTTTGAA, GAGGTAAGCCTACACTTTCCAAGA

ITGB1, CCGCGCGGAAAAGATGAAT, CCACAATTTGGCCCTGCTTG”

(Supplementary Material)

・The following points were added to Materials and Methods (Target gene knockdown using RNA Interference section)

“SOX18 mRNA levels were evaluated for each condition at the same time. The mRNA expression levels of SOX18, as well as VCAM1, ICAM1, ITGB1, and IL-6, were evaluated for each condition.” (L.172-173)

“Effects of S100A6, HSP90AA1, and TXNIP knockdown on the expression of PAH disease-related genes. Relative mRNA expression levels of VCAM1, ICAM1, ITGB1, and IL-6 were analyzed by RT-qPCR in control and knockdown (KD) cells. Expression levels were normalized to GAPDH. Graphs are presented as median with interquartile range (IQR). Statistical significance is indicated as: ** p < 0.01, ns: not significant (n = 4 for Control; n = 6 for KD).” (Legend of Fig. S2)

[The second half]

To validate our findings, we utilized the independent single-cell RNA-seq dataset GSE228644. This analysis confirmed a significant upregulation of SOX18 in PAH endothelial cells (Log2FC: 0.85). Furthermore, to ensure this change represents a true biological signal rather than natural variance, we performed a permutation analysis using the healthy donor cells (n = 1,307) from this dataset. We established a natural fluctuation threshold at the 95th percentile (± 0.68) based on 100 random split iterations. The observed SOX18 upregulation significantly exceeded this noise threshold by 1.24-fold (Mann-Whitney U test, p = 1.31 × 10⁻⁴) (Fig.S1 A, B). These results provide robust external validation of our experimental data, demonstrating that SOX18 upregulation is a distinct pathological feature that lies well beyond the range of natural fluctuations.

Based on the above experimental results and discussions, the following description, corresponding legends have been added to the text.

・The following points were added to Materials and Methods (we newly added “Estimation of natural fluctuation threshold” section.)

“To distinguish true biological signals from technical noise or natural biological variance, we performed a permutation-based analysis using the control samples from the validation dataset GSE228644. We randomly split the 1,307 healthy donor cells into two equal groups 100 times and calculated the log2 fold change (log2FC) for all genes in each iteration. From the resulting distribution of background log2 FC values, we defined the "natural fluctuation zone" as the range covering the 95th percentile. An observed log2FC in the disease group exceeding this threshold was considered a robust change beyond natural variance.” (L.186-194)

・The following points were added to Results.

“To validate our findings, we utilized an independent scRNA-seq dataset (GSE228644). This analysis confirmed a significant upregulation of SOX18 in PAH endothelial cells compared to donors (log2FC = 0.85). To address the potential for natural variation, we established a natural fluctuation threshold (log2FC ± 0.68) based on a permutation analysis of healthy donor cells. The upregulation of SOX18 significantly exceeded this noise threshold (1.24-fold above the threshold limit), confirming that the observed expression change represents a distinct pathological feature rather than intrinsic biological noise.“

“External validation of SOX18 upregulation using an independent dataset.

(A) Null distribution versus actual SOX18 change: Permutation analysis defining the natural fluctuation of gene expression. The density plot shows the distribution of Log2 fold changes derived from healthy donor endothelial cells (n = 1,307) in the GSE228644 dataset across 100 random split iterations. The orange shaded region represents the natural fluctuation zone, defined as the 95th percentile (log2 FC = 0.68). The red arrow indicates the observed Log2 fold change of SOX18 in PAH versus donor cells (log2 FC = 0.846).

(B) SOX18 expression in endothelial cells: Violin plot comparing SOX18 expression levels between healthy donors and PAH patients in the GSE228644 dataset (Mann-Whitney U test, P = 1.31×10⁻⁴).” (Legend of Fig. S1)

The source code for PAH-former is available at the following repository https://github.com/UTcardiology/PAH-former-analysis. Custom scripts used for the permutation analysis and natural fluctuation threshold estimation are also available in the same repository.

Major Critique: There are many genes now identified perhaps the authors should compare vs genes with known targeted therapeutics.

Response:

Following this suggestion, we conducted in silico perturbation analyses on well-established PAH therapeutic targets, including Phosphodiesterase 5A (PDE5A), soluble guanylyl cyclase stimulators (GUCY1A3, GUCY1B3), endothelin receptors (EDNRA, EDNRB), and prostacyclin (PTGIR).

We tested four perturbation directions and we have attached an Excel file containing the "Shift_to_goal_end" scores for each gene, calculated via in silico perturbation analysis using PAH-former in this revision. We found that perturbation of these specific receptor/enzyme genes alone did not induce a statistically significant global state transition (FDR > 0.5) in our model. As mentioned in our manuscript, "Shift_to_goal_end" represents the cosine similarity shift of the cell embedding vector towards the centroid of the target state (healthy or disease) following the in silico perturbation of a specific gene. A higher value indicates a stronger predicted impact driving the cell toward the target state. This result likely reflects that while these targets are clinically effective for vasodilation and symptom management, modulating their individual mRNA expression levels may not be sufficient to drive a global reversion of the complex transcriptomic dysregulation characteristic of PAH in this model.

However, when we focused on the Deletion: PAH to Control direction—which simulates the therapeutic inhibition of pathogenic genes—our model successfully prioritized several previously reported driver genes of PAH, such as the AP-1 complex, TGF-β (a downstream effector of PPAR-γ signaling), and SPARC (a target of HIF-2α) (5). In the pathway analysis of genes extracted from the "Deletion: PAH to Control" perturbation, the PID-AP1 pathway was also identified.

These findings suggest that while PAH-former may not prioritize every target genes of current vasodilators, it is highly effective at identifying upstream transcriptional regulators required to normalize the global transcriptomic state. Therefore, our strategy provides a reliable method for finding new treatment targets that address the fundamental molecular pathology of PAH.

Summary of in silico perturbation scores for established PAH therapeutic target genes (Extracted from supplementary table 1)

Based on the above experimental results and discussions, the paper was revised as follows.

・The following points were added to Discussion.

“Furthermore, to benchmark our model against established clinical targets, we performed in silico perturbation analyses on genes targeted by current PAH therapies, such as Phosphodiesterase 5A (PDE5A), Guanylate Cyclase 1 Soluble Subunit Alpha 3 (GUCY1A3), Endothelin Receptor Type A (EDNRA), and Prostaglandin I2 Receptor (PTGIR). Interestingly, perturbation of these individual genes did not induce a significant global shift in cell embedding towards the healthy state. While these targets are clinically effective for vasodilation and symptom management, modulating their expression alone may not be sufficient to reverse the complex, global transcriptomic dysregulation characteristic of the PAH endothelial phenotype. In contrast, when we focused on the in silico deletion analysis simulating the reversion from the PAH state to the control phenotype—which simulates the therapeutic inhibition of pathogenic genes—our model successfully prioritized major upstream regulators and signaling nodes, such as the Activator Protein 1 (AP-1) complex, Transforming Growth Factor Beta (TGF-β), and Secreted Protein Acidic and Cysteine Rich (SPARC) which are well-known key factors involved in the pathophysiology of PAH (5). This suggests that PAH-former is particularly effective at identifying fundamental drivers required to normalize the global cellular state.” (L.347-363)

Major Critique: the figures are not of high quality; some of the data is illegible.

Response:

We addressed your comment.

Comments from Reviewer #2

Major Comments: For the target gene knockdown analysis section, the authors mentioned picking four genes for further validation. It is not clear what the rationale was behind choosing these genes. Is it data-driven or biology driven? This needs to be clarified or cleared up in the manuscript.

Response:

The identification of the four genes for validation was guided by a combined approach that prioritized data-driven predictions while incorporating biological context and experimental feasibility. First, we identified top-ranked genes with high control to PAH shift scores, which is previously mentioned as “Shift_to_goal_end”. Next, we carefully examined each of these top candidates' biological functions one by one. In consultation with the PAH experts among the co-authors, we carefully chose genes with functional traits that clearly suggested a possible connection to the pathophysiology of PAH, even though they had not previously been directly reported in the previous literature. Finally, we confirmed that these candidates showed detectable basal expression in HPAECs based on public transcriptomic data (GSE228644) to ensure experimental feasibility.

・The following points were added to Materials and Methods (Target Gene Knockdown using RNA Interference section). The term ‘HPAECs’ is spelled out at its first mention.

“To choose targets for experimental validation, we used a multi-step procedure. Top-ranked genes discovered by Control-to

---

## [Decision Letter · Decision Letter 1]

16 Feb 2026

PAH-former: Transfer learning for efficient discovery of pulmonary arterial hypertension-associated genes

PONE-D-25-64558R1

Dear Dr. Ito,

We’re pleased to inform you that your manuscript has been judged scientifically suitable for publication and will be formally accepted for publication once it meets all outstanding technical requirements.

Kind regards,

Nanako Kawaguchi

Academic Editor

PLOS One

Additional Editor Comments (optional):

The revised ms is improved. I found one typo in Line 219. Genformer should be corrected to Geneformer.

Reviewers' comments:

Reviewer's Responses to Questions

**Comments to the Author**

Reviewer #1: All comments have been addressed

Reviewer #2: All comments have been addressed

2. Is the manuscript technically sound, and do the data support the conclusions?

Reviewer #1: Yes

Reviewer #2: Yes

3. Has the statistical analysis been performed appropriately and rigorously?

Reviewer #1: Yes

Reviewer #2: Yes

4. Have the authors made all data underlying the findings in their manuscript fully available?

Reviewer #1: Yes

Reviewer #2: Yes

5. Is the manuscript presented in an intelligible fashion and written in standard English?

Reviewer #1: Yes

Reviewer #2: Yes

Reviewer #1: All comments were addressed by the authors. The added methodology and analyses have strengthened the paper. The manuscipt is ready for publication.

Reviewer #2: (No Response)

**Do you want your identity to be public for this peer review?** For information about this choice, including consent withdrawal, please see our Privacy Policy

Reviewer #1: No

Reviewer #2: **Yes:** Sudhir Ghandikota

---

## [Editor Report · Acceptance letter]

PONE-D-25-64558R1

PLOS One

Dear Dr. Ito,

I'm pleased to inform you that your manuscript has been deemed suitable for publication in PLOS One. Congratulations! Your manuscript is now being handed over to our production team.

Kind regards,

on behalf of

Dr. Nanako Kawaguchi

Academic Editor

PLOS One